# A Comparative Experimental and Computational Study on the Nature of the Pangolin-CoV and COVID-19 Omicron

**DOI:** 10.3390/ijms25147537

**Published:** 2024-07-09

**Authors:** Lai Wei, Lihua Song, A. Keith Dunker, James A. Foster, Vladimir N. Uversky, Gerard Kian-Meng Goh

**Affiliations:** 1College of Life Science and Technology, Beijing University of Chemical Technology, Beijing 100089, China; weilai.chongqing@gmail.com; 2Center for Computational Biology and Bioinformatics, Department of Biochemistry and Molecular Biology, Indiana University School of Medicine, Indianapolis, IN 46202, USA; kedunker@iu.edu; 3Department of Biological Sciences, University of Idaho, Moscow, ID 83844, USA; foster@uidaho.edu; 4Institute for Bioinformatics and Evolutionary Studies, University of Idaho, Moscow, ID 83844, USA; 5Department of Molecular Medicine, Morsani College of Medicine, University of South Florida, Tampa, FL 33612, USA; vuversky@health.usf.edu; 6Goh’s BioComputing, Singapore 548957, Singapore

**Keywords:** coronavirus, COVID, intrinsic disorder, membrane, nucleocapsid, nucleoprotein, omicron, pangolin, shell, virulence, long COVID, delta, attenuation, virulence, variant, XBB, AI, artificial intelligence, vaccine

## Abstract

The relationship between pangolin-CoV and SARS-CoV-2 has been a subject of debate. Further evidence of a special relationship between the two viruses can be found by the fact that all known COVID-19 viruses have an abnormally hard outer shell (low M disorder, i.e., low content of intrinsically disordered residues in the membrane (M) protein) that so far has been found in CoVs associated with burrowing animals, such as rabbits and pangolins, in which transmission involves virus remaining in buried feces for a long time. While a hard outer shell is necessary for viral survival, a harder inner shell could also help. For this reason, the N disorder range of pangolin-CoVs, not bat-CoVs, more closely matches that of SARS-CoV-2, especially when Omicron is included. The low N disorder (i.e., low content of intrinsically disordered residues in the nucleocapsid (N) protein), first observed in pangolin-CoV-2017 and later in Omicron, is associated with attenuation according to the Shell-Disorder Model. Our experimental study revealed that pangolin-CoV-2017 and SARS-CoV-2 Omicron (XBB.1.16 subvariant) show similar attenuations with respect to viral growth and plaque formation. Subtle differences have been observed that are consistent with disorder-centric computational analysis.

## 1. Introduction

### 1.1. COVID-19-Related Viruses

SARS-CoV-2 was first observed among patients in November 2019 in Wuhan, China [1,2,3,4]. Its sequence was quickly determined, and SARS-CoV-2 was found to have approximately 80% genetic similarity with the 2003 SARS-CoV (SARS-CoV-1). A search for closely related CoVs pointed to a 2013 sample (bat-RaTG13) obtained in a cave in Yunnan that had a 96.4% sequence identity with SARS-CoV-2 [5,6]. Interestingly, two sets of pangolin-CoV samples obtained were found to have approximately 90% similarities to SARS-CoV-2 [7,8,9]. One set was obtained from smuggled pangolins confiscated in Guangdong (GD) province in 2019. The other samples were also from smuggled pangolins but from Guangxi (GX) province (2017–2018) [7,8,9,10]. Later, similar data were found in bat and pangolin samples from Laos and Vietnam, respectively [11,12]. The series of CoVs found among bats in Laos were labeled BANAL, with one of the samples (BANAL-52) possessing a 96.8% genetic proximity to SARS-CoV-2 that surpasses RaTG13 [3,11].

The computational models used in this paper, Shell Disorder Models (SDMs), were built before the COVID-19 emergence. SDMs used AI (Neural Network) [13,14] that measures levels of disordered residues in two CoV shell proteins, membrane (M) and nucleocapsid (N) [14]. A specific characteristic of the COVID-19-related viruses is that they all have abnormally hard outer shells (i.e., are characterized by low PID (Percentage of Intrinsi Disorder) values of their M proteins, PID_M_), which are found to be associated with CoVs of burrowing animals, such as pangolins and rabbits [15,16]. Viruses with hard outer shells, such as dengue and rabies viruses, are often associated with saliva that has powerful antimicrobial enzymes that could damage the virion [14,15,16,17,18]. It is believed that the reason for the high contagiousness of SARS-CoV-2 has to do with the ability of the virus to resist the onslaught of salivary antimicrobial enzymes as a result of its hard outer shell, which allows a large amount of virus shedding by patients [19]. This paradigm provides an important potential evolutionary connection between SARS-CoV-2 and the pangolin, which is a burrowing animal. Viral transmission of burrowing animals is likely to involve feces that are buried for a long time and therefore necessitate the transmitted virus to persist in the ground, for which a hard outer shell is needed [15,16,19].

The N protein, on the other hand, plays important roles in the replication process, and thus greater N disorder provides the means necessary for more rapid and efficient replication [14]. It is for this reason that greater disorder at the inner shell has been found to correlate with greater virulence in many viruses, such as DENV, Ebola virus (EBOV), SARS-CoV-1/2, and Nipah virus (NiV) [14,15,16,17,18,19,20]. This prediction represents a part of the Shell Disorder Model (SDM). Upon inspection of the PID_N_ of pangolin-CoVs, a larger range of PID_N_ values is observed. Most interestingly, according to SDM, the pangolin-CoV 2017 [15,19] is predicted to be attenuated, especially when compared to the Wuhan SARS-CoV-2 strain (Wuhan-Hu-1). Some of the co-authors of this paper published this observation in 2020 [15]. Later, the other co-authors published papers independently of the aforementioned 2020 computational study, which experimentally showed that Pang2017 (GX_P2V) is attenuated [21,22]. Similar characteristics were later found in the case of SARS-CoV-2 Omicron [23,24,25,26,27]. Intriguingly, the GD pangolin-CoV from 2019 (Pang2019) did not show any such attenuation experimentally [28]. Similar results involving greater aggressiveness can be found in some of the bat-CoVs found in Laos, particularly BANAL-236 [11]. Even as puzzling as these results are to some researchers, all these results did not come as a surprise to many of the co-authors, as the SDMs had consistently predicted all these peculiarities.

### 1.2. Peculiar Relationships between SARS-CoV-2, Omicron, and Pangolin-CoVs

Since the first discovery of SARS-CoV-2 in 2019 (Wuhan-Hu-1), SARS-CoV-2 has undergone extensive mutagenesis. The variants are identified by mutations found at the S protein and given labels such as Alpha and Beta [29,30,31,32]. In November 2021, the Omicron variant was first detected in South Africa. Omicron is enigmatic by itself, as the mutations found in the virus are unlike any other known variants [26]. These observations raised important questions, such as: where was the direct Omicron ancestor hiding all along? Why was it previously never noticed in the human population? Another characteristic of Omicron has to do with its attenuated nature. This observation was made very early in the spread in South Africa, when physicians observed relatively mild symptoms among patients infected by Omicron, especially when compared to the patients infected by the previous SARS-CoV-2 variants [29,30,31]. Upon inspection of the shell disorder, we were able to reproduce the mentioned observation of attenuation by detecting that the Omicron PID_N_ was similar to that of Pang2017. A group of us published the corresponding results in 2022 [19,32]. The present study examines the computational and experimental data that are associated with the peculiar relationships among pangolin-CoV, non-Omicron SARS-CoV-2, and Omicron.

## 2. Results

### 2.1. The Shell Disorder Models (SDMs)

The SDMs are a series of closely related models that measure the disorder propensity of the viral shell proteins in order to analyze and understand the behaviors of the viruses, as seen in Table 1. The parent model is the “Shapeshifter” model that was published in 2008, when it was the first to report the presence of an abnormally disordered outer shell (matrix) among a large number of HIV-1 variants [14,17]. We attributed the unsuccessful search for an HIV vaccine to this unique property, seldom seen among viruses. Another model was published in 2015 [14,18,20]. This model showed that there are strong correlations between disorder at the inner shell and the virulence of many viruses, including DENV, EBOV, and NiV [14,15,16,17,18,20,27] We shall see that it applies also to COVID-19-related viruses. The reason for the correlation between greater inner shell disorder (PID_N_, the percentage of intrinsically disordered residues in the N protein) and virulence has to do with the important roles that the inner shell proteins often play in the replication of the viruses [27,28,33,34,35,36,37,38,39]. The replication processes require the binding of the inner shell proteins to RNA/DNA and other viral and host proteins. The greater disorder provides means for more efficient binding and, therefore, faster replication. Many viruses take advantage of their ability to replicate more rapidly as a form of immune evasion. This “Trojan-horse” immune evasion involves the rapid replication of the virus via shell disorder even before the host immune system recognizes its presence. This strategy often, however, backfires on the virus by overwhelming the host body with abundant copies of infectious particles, which kill the host [14,27,37].

In the case of CoVs, N plays a distinct role in the replication of the virus. An important role is the packaging of the infectious particles prior to their release. N has to wrap itself around the RNA, and the greater N disorder provides for more efficient RNA-binding and, thus, more efficient assembly of the infectious particles that will later be released [40,41]. A second mode of action that N plays in the infectivity of the virus involves the entry of N into the nucleus of the cell in order to manipulate the cell cycle to the advantage of the virus. Greater N disorder could provide greater efficiency in its entry into the nucleus since greater disorder could enhance protein-protein and protein-lipid interactions [27,28,33,34,35,36,37,38,39]. The domains in N that are responsible for the two modes of action will be discussed in greater detail later in this paper.

In 2012, the third SDM was published [42]. This is the CoV Transmission Model, in which levels of fecal-oral and respiratory transmissions were correlated to the levels of disorder of the shell proteins. Initially, three groups of CoVs were categorized. Their correlations with the modes of transmission were established using clinical and epidemiological data from farm animals, particularly porcine-CoVs. We need to keep in mind that much of the data available before 2003 were from the veterinary community, not the medical community, as CoVs were mainly associated with mild symptoms among humans before 2003, and they were not considered medically interesting [14,42] then. With the arrival of SARS-CoV-2, a fourth group that is associated with burrowing animals was detected and added. Figure 1 describes the four groups. CoVs in group A have higher PID_N_ values, which allow for higher respiratory transmission potentials. As aforementioned, the inner shell protein, N, is intimately involved in the replication process, and a higher disorder of this protein (greater PID_N_) allows for more efficient interactions between N and the RNA and other proteins [4,5,6,7,8,9,10,11,12,13,14,14,15,16,16,17,18,19,19,20,21,22,23,24,25,26,27,28,29,30,31,32,33,34,35,36,37,38,39,40,41,43]. This greater efficiency leads to higher shedding of the virus that is necessary for respiratory transmission. The three groups, A–C, were categorized according to this principle, as seen in Figure 1 [14]. Viruses in group A have the highest PID_N_ and have higher respiratory transmission potentials but lower fecal-oral potentials. Conversely, the CoVs in group C have higher fecal-oral but lower respiratory transmissibility potentials. CoVs in group B, on the other hand, have intermediate levels of both fecal-oral and respiratory transmissibility potentials.

The initial model comprises groups A, B, and C (see Figure 1), which are mainly based on PID_N_, and, in fact, a strong correlation could be found between PID_N_ and the mode of transmission. M, on the other hand, poses an enigma at that time. A small but statistically significant correlation between PID_M_ and mode of transmission could be found [16,43]. The precise significance of this correlation was difficult to construe at that time. It was not until the arrival of COVID-19 with its incoming data that we fully realized that the reason for our inability to further analyze the significance of PID_M_ then was due to the scarcity of data pertaining to CoVs with extremely low PID_M_ that are associated with burrowing animals [16,19,32,42].

A fourth category, group D, was discovered when data pertaining to COVID-19 poured in [19,32,37]. This group consists of CoVs that have abnormally hard outer shells (i.e., their M proteins are characterized by low disorder content and low PID_M_) that are usually associated with a burrowing animal host, such as rabbits and pangolins. The association between burrowing animals and the hard outer shell of the virus has to do with the necessity for the virus to persist in feces that could be buried for a long time until encountered by another host [15,16,19,32]. It is believed that CoVs in this category have both high fecal-oral and respiratory transmission potentials. Because of their hard outer shells, the viruses are able to resist the onslaught of antimicrobial enzymes found in the saliva and mucus [44,45,46,47,48]. As in the case of COVID-19, the infected host body typically sheds large amounts of infectious particles, unlike SARS-CoV-1 [49]. Keeping in mind that COVID-19-related viruses are considered somewhat unique in their extremely hard outer shell [15,16,19,32], we believe that this feature, being associated with the peculiarities of the physiological properties of the mammalian respiratory system (Mucociliary Clearance System) [50,51,52], is responsible for the high infectivity of the virus [19,32].

The three SDMs provide specific predictions, which are highly reproducible thus far. For instance, SARS-CoV-1 is predicted to be more virulent but less transmissible than SARS-CoV-2. SARS-CoV-1 has a PID_N_ of 50% and a PID_M_ of 9%, whereas SARS-CoV-2 (Wuhan-Hu-1) has 48.2% PID_N_ and 5.9% PID_M_. The percentage difference between 50% and 48.2 can be very misleading, as N is a huge protein with over 400 amino acid residues and is a highly abundant protein both in the virion and cell [38,39]. Therefore, even a small difference in percentage could entail a fairly large number of mutations that involve widespread regions of disorder. This is also the case with M, as M is somewhat smaller but is the most abundant protein in the virion [39,42]. The Virulence-SDM predicts that SARS-CoV-1 will be more pathogenic based on its higher PID_N_, and this has been supported by the fact that the Case Fatality Rate (CFR) of SARS-CoV-1 is about 10%, while the Wuhan-Hu-1 was estimated to have a CFR of around 2%. The greater N disorder allows for more efficient replication of SARS-CoV-1, especially in vital organs, whereas the harder outer shell (lower PID_M_) of SARS-CoV-2 causes greater shedding by the patients. The much higher level of shedding by COVID-19 patients has also been clinically observed [49].

We have now seen that greater disorder in the N (inner shell) can bring about greater virulence or greater infectivity, depending on many factors, such as the viral presence in vital organs, saliva, or mucus. This idea is consistent not just in CoVs but also in other viruses we have studied in the past, when it was found that the inner shell of unrelated viruses such as DENV, EBOV, and NiV [14,17,18,20,27,32], correlates with virulence. The outer shell (M, in the case of CoVs) is, however, more enigmatic. We don’t have any evidence that the inner shell disorder is correlated with the mortality rate of any virus. We have, however, some hints that the outer shell can be related to the virus’ ability to penetrate vital organs and cause damage, i.e., morbidity, not mortality, in cases such as Yellow Fever virus, Zika virus and HIV [14,17]. As already mentioned, in the case of CoVs, we have discovered that the hardness of M is related to the infectivity of COVID-19. We will also revisit this in Figure 2.

Figure 2A summarizes how SARS-CoV-2 is able to shed more infectious particles yet remain relatively less virulent than SARS-CoV-1. It produces fewer copies in vital organs, but because of its hard outer shell, the virus is able to resist the anti-microbial enzymes and thus shed more particles, unlike SARS-CoV-1. Figure 2B illustrates the strong relationship between COVID-19 and pangolin. Intermediary animal hosts often provide a venue for attenuation, as in the case of pigs in the Malaysian Nipah outbreak in 1999–2000, because of the behavior of the animals [14,27,53]. Likewise, SARS-CoV-2 was more efficiently attenuated because it is likely to have been passed between humans and pangolins for at least several years before the appearance of Wuhan-Hu-1 [15]. This is likely also the case in Omicron [19,32]. Further evidence of interspecies transmissions can be found in the facts that SARS-CoV-2 is capable of infecting a large number of mammalian species [54] and that bats often dwell alongside pangolins in caves and underground burrows [55].

### 2.2. Phylogenetic Study Using M Reveals Intimate Relationship between Pangolin-CoV and SARS-CoV-2/Omicron

We have seen that a common characteristic of all thus far known COVID-19-related viruses is the abnormally hard outer shell (low PID_M_), with the high rigidity of M being tied to the high COVID-19 infectivity and the evolutionary close relationship between SARS-CoV-2 and a burrowing animal, the pangolin [15,19,32]. We have also seen that Omicron presented the enigma of being different from all previous SARS-CoV-2 variants, which may suggest that Omicron had been hiding in an animal host. The phylogenetic tree using M shown in Figure 3 provides evidence to support this scenario by grouping both Omicron XBB and BA2 closer to pangolin-CoVs and bat-CoVs than the other SARS-CoV-2 variants [19,32]. A previous study using M and different software and algorithms involving distance optimization resulted in an even closer relationship between Omicron and pangolins. The phylogenetic studies using M are the likeliest to present the most accurate trees because M needs to be highly conserved given its rigidity. A highly conserved protein provides greater accuracy in phylogenetic studies since studies have shown that current phylogenetic algorithms often handle recombination poorly [56]. In fact, previous phylogenetic studies using M were shown to offer a unique perspective on the evolution of SARS-CoV-2 as related to pangolin-CoVs in comparison to other studies.

### 2.3. Omicron and Pang2017: Low PID_N_ and Attenuation

Evidence of SARS-CoV-2 attenuation by its association with pangolins can be found in Table 2. Most of the bat-CoVs have higher PID_N_ values (48.5%) than Wuhan-Hu-1 (48.2%). The pangolin-CoVs have a much wider range of PID_N_ values, which suggests that pangolins may provide a suitable environment for the attenuation of the virus since a harder inner shell does also provide some further protection to the virion [14,18,32]. We also see in Table 2 that the range of PID_N_ values of SARS-CoV-2, including Omicron resembles more closely those of pangolin-CoVs than those of bat-CoVs, even if the latter is genetically closer.

### 2.4. Omicron Has a Lower PID_N_ Similar to Pango 2017 but Has a Lower PID_M_: Attenuation and Faster Spread

Further evidence that levels of N disorder have a strong correlation (r = 0.9, *p* < 0.05) with virulence can be found in Figure 4, where a clear Omicron attenuation is seen. The underlying theoretical link is found in the Virulence-Inner SDM, in which the greater disorder at the inner shell protein allows faster replication of the virus, which often leads to higher virulence. Because CFRs of SARS-CoV-1 and SARS-CoV-2 variants are available, the correlation between the PID_N_ values and virulence can be calculated. More data became available when CFR (<0.2%) of patients, especially non-vaccinated ones, infected by Omicrons became available [26,34]. This can be compared to the other earlier SARS-CoV-2 variants (2–0.5%) [23,24,25,29,30,31,43]. Even though Figure 4 includes pangolin-CoV and bat-CoV, we do not really know the CFR of these viruses. However, Figure 4 makes predictions on the likely CFR of such viruses if they infect the human population, and, indeed, as we will see later, the PID_N_ is found to correlate with the aggressiveness of the virus under experimental conditions.

### 2.5. The Role of N in CoV-Transmission SDM and Virulence-Inner Shell Disorder Model

Based on the available data for SARS-CoV-1 and SARS-CoV-2 variants, a correlation between PID_N_ values and virulence can be calculated. This analysis revealed a strong correlation (r = 0.8). This is in sharp contrast to Figure 1, where PID_N_ has a strong correlation to the mode of transmission, especially in groups A–C. The two correlations are based on the same theoretical principle. The Virulence-Inner SDM, as seen in Figure 4, entails higher PID_N_ for greater pathogenesis of the virus because higher disorder levels in N facilitate faster replication of the virus through greater protein-protein/RNA/Lipid binding efficiency [36,37,38,39] and large amounts of a virus in vital organs often kill the host [14,18,19,20,32,59,60].

### 2.6. All Known COVID-19 Viruses a Have Hard Outer Shell: Evolutionary Association with Pangolins

While it is easy to assume that bat-CoVs, such as RaTG13 and BANAL-52, are most closely related to SARS-CoV-2, such an assumption could be misplaced as many other considerations, such as recombination, are involved. There are also evolutionary patterns that we need to be concerned about. Several papers have already noted that parts of the S protein of pangolin-CoVs most closely resemble those of SAR-CoV-2 [61]. We have also seen that the range of PID_N_ values of pangolin-CoVs resembles that of SARS-CoV-2, especially when Omicron is included. There is yet another mystery that involves the origin of Omicron. When Omicron first emerged, it was already noted that it did not resemble any other previous variant, and the types of mutations found in it had previously never been seen. If so, where was Omicron hiding? The phylogenetic tree using M in Figure 3 offers some clues by showing the surprisingly close relationship between pangolin-CoVs and Omicron. M is likelier to offer greater reliability in phylogenetic studies of COVID-19-related viruses. In fact, as we have seen in Figure 1 and Table 2, all COVID-19 CoVs have abnormally hard M, which implies that M is likelier to be the most highly conserved protein, which makes it the best choice in such studies. Figure 3 shows that Omicron (BA.1.44 and XBB.1.16) are clustered together.

### 2.7. Correlation between Viral Growth and N Disorder of COVID-19-Related Viruses

We noticed that, in Figure 4, the CFRs for pangolin-CoVs and bat-CoVs are not available, as human infections have not been observed for those viruses even though laboratory studies have shown that they are capable of infecting a wide range of animals, including humans [54]. Given the strong correlation between CFRs and virulence, the SDMs predict that the aggressiveness of the various COVID-19-related viruses will correlate with the PID_N_ values. This is because the SDMs involve the paradigm that greater N disorder allows for more rapid replication of virus copies, which leads to greater virulence. An examination of experiments by other laboratories and our associated laboratory provides evidence of a correlation (r = 8) between the aggressiveness of the virus and PID_N_ levels, as seen in Table 3.

Hou et al. [41] showed that under certain circumstances, Pang2019 replicates more efficiently in respiratory cells than SARS-CoV-2. Temman et al. [11] showed that BANAL-236 has about the same replication rates as SARS-CoV-2 when Vero-E6 or Caco-3 cells are used. Ogando et al. [62] demonstrated that SARS-CoV-1 replicates fifty times faster than SARS-CoV-2 in Vero-E6 cells. Guo et al. [21] and Lu et al. [22] have found that Pang2017 is attenuated, especially when compared to non-Omicron SARS-CoV-2 variants. As for Omicron, numerous clinical and experimental studies [29,30,31,40] have shown it to be milder than other variants. The viral growths of XBB.1.16 and Pang2017 are addressed later in this paper. All these data were used to construct Table 3.

The repetitive interspecies transmission may provide insight into the mechanisms by which SARS-CoV-2 acquires its unusual multi-species adaptation [53]. It also reiterates the suggestion made in our previous publications that an attenuated COVID-19-related virus may have entered the human population in 2017 or earlier. An entry of an attenuated virus could easily have been mistaken for a mild cold by the medical community, especially if the spread is slow.

### 2.8. Molecular Analysis SARS-CoV-2’s Evolution within Animals Affects Its Virulence and Human Spread

Results of more detailed computational analysis of N proteins from SARS-CoV-1, Wuhan-Hu-1, Omicron XBB.1.16, and pangolin-CoV-2017 can be found in Figure 5. Figure 5A represents the PONDR^®^ VLXT-generated intrinsic disorder profiles of these proteins, where the X-axis represents the locations of the residues and the Y-axis shows the corresponding PONDR^®^ VLXT scores. Regions with scores of 0.5 and above are considered to be intrinsically disordered regions.

While Figure 4 and Table 2 tell us that the virulence of SARS-CoV-1, Wuhan-Hu-1, Omicron, and Pang2017 is related to the levels of their respective PID_N_ values, Figure 5 points out the exact locations of the differences in the N disorder. As seen in Figure 5A,B, the NTD (N-terminal domain) responsible for the RNA binding by the N protein is located within the 44-186 region, the dimerization domain occupies the 258-361 region, whereas the NLS (Nuclear Localization Signal) motif lies in the 366-381 region [19,37,38,39,42]. While the viral RNA binds to other regions of the N protein, the NTD RNA-binding domain is arguably the main RNA-binding region of this protein. We know from Figure 4 that SARS-CoV-1 is more virulent than Wuhan-Hu-1, as the former is associated with higher PID_N_. Figure 5B,C shows that the higher levels of the intrinsic disorder can be found both inside and outside the NTD RNA-binding region of the Wuhan-Hu-1 N protein. The N protein is found both within the cytoplasm and nucleus, where it could manipulate the cell cycle to the virus’s advantage, and the NLS site provides means for this protein to enter the nucleus. Importantly, the region encompassing this motif is predicted to be highly disordered in all four proteins analyzed in this study.

A comparison of disorder profiles in Figure 5A suggests that while the disorder differences between SARS-CoV-1 and Wuhan-Hu-1 can be found in regions at or near the NLS, NTD RNA-binding domain, and dimerization domain, the differences between Wuhan-Hu-1 and Pang2017 lie mainly in and around their NTD RNA-binding domain but not in other regions. This implies that the attenuation of Pang2017 is likely to arise from the modulation of protein-RNA binding, not protein-protein or protein-membrane interactions, which provide us with ideas on ways to further attenuate Pang2017. Figure 5B allows tracing the source of the discrepancies in disorder by sequence comparison. There is a large ordered region in the Pang2017 NTD (residues 14–23), not seen in Wuhan-Hu-1 (Figure 5A). It should be noted that even though this ordered region is not inside the NTD RNA-binding domain (i.e., residues 44–187), it is adjacent to it and is therefore likely to affect the binding potential.

The residues responsible for this disorder discrepancy can be seen in Figure 5B. At position 9, a mutation took place involving the replacement of an R (Arginine) by a Q (Glutamine) in the case of SARS-CoV-2. This point mutation is followed by two insertions of R (Arginine) and N (Asparagine). Therefore, instead of one disorder-promoting residue as seen in Pang2017, this region of the SARS-CoV-2 N protein has three disorder-promoting residues.

An important point to be made in this paper is that SDMs have determined that Pang2017 and Omicron have similar levels of attenuation based on their PID_N_ values. Even though the viruses have similar levels of N disorder, slight differences can be detected depending on the specific Omicron subvariant used. Two Omicron subvariants, BA.1 and XBB.1.16, are used, but XBB.1.16 is our point of reference both computationally and experimentally. As seen in Table 2, BA.1.44 and XBB.1.16 have PID_N_ values of 44.7% and 44.2%, respectively, whereas Pang2017 PID_N_ is 44.8%. The small difference between the PID_N_ values of XBB.1.16 and Pang2017 could explain the subtle but perhaps important experimental discrepancy between their viral growth in Vero cell as we shall see later. A comparison of the PONDR^®^ VLXT plots in Figure 5 reveals two regions responsible for much of the mentioned discrepancy. One region sits at the NTD RNA-binding domain, whereas the other involves the NLS domain. The site at the NLS domain responsible for much of the discrepancy lies near position 384, where Pang2017 has slightly greater disorder. Figure 5C shows the mutations 374TA373, 377ST376, and 379PA378. The mutations generally involve the replacement of a more polar residue with a less polar one, as in the case of Pang2017. Polar residues tend to cause greater disorder. The existence of higher local disorder, even if the difference is not very large, at or near the NLS is likely to indicate that Pang2017 has a slightly greater ability to penetrate the nuclear of the host cell during replication, as greater disorder provides means for better protein-protein interactions and, thus, allows some added advantage during replication.

A different story can be found at the NTD RNA-binding domain. Despite the fact that pang2017 N is generally slightly more disordered, XBB.1.16 is seen as having greater disorder around location 70 (Figure 5). This greater local disorder level may allow XBB.1.16 to have higher RNA-binding efficiency, which is likely to be reflected in our experimental part, as we will see later. An examination of the sequences using Figure 5C tells us that this greater disorder is likely arising from a sequence insertion and deletion at locations 28 and 9, respectively, for Pang2017.

The introduction of SARS-CoV-1 into humans may be linked to an intermediary involving Civet cats. Because of the higher virulence of SARS-CoV-1 in comparison to SARS-CoV-2 and the high genetic similarity of SARS-CoV-1 and Civet-CoV, it can be suspected that SARS-CoV-1 entered civets for a short while before moving to humans. The lower virulence of SARS-CoV-2 leads us to believe that the virus may have been incubating in an intermediary involving a burrowing animal, such as pangolin, for a relatively longer period of time. The case of NiV illustrates this point. The virus involved in the 1999–2000 Malaysian outbreak is less virulent than the viruses involved in outbreaks in Bangladesh and India because the former involved an intermediary, i.e., farm pigs.

### 2.9. Comparison of Cytopathic Effects, One-Step Growth Curve, and Plaque Size of Pang2017 and SARS-CoV-2 XBB.1.16 in Vero Cells

Vero cells were infected with Pang2017 and XBB.1.16 at an MOI of 0.1. At 36 h post-infection, no notable changes were observed in cells infected with Pang2017, while those infected with XBB.1.16 exhibited pronounced cytopathic effects. By 48 h post-infection, cells infected by both viruses began to round up, shrink, and detach. Between 48 and 72 h post-infection, a greater number of cytopathic cells were observed in the Pang2017-infected group, but the monolayer density was considerably higher than in cells infected with XBB.1.16.

Using qPCR to measure the viral RNA load in the culture supernatant, the one-step growth curves of the two viruses were determined. Overall, both viruses displayed similar growth curves, with their log phase spanning between 12–36 h. Between 8 and 10 h post-infection, the RNA copy number of Pang2017 rose by approximately one log, while there was no significant change in the XBB.1.16 RNA copy number. From 10 to 48 h post-infection, the Pang2017 RNA load in the supernatant was consistently higher than that of XBB.1.16. By 48 h post-infection, the RNA loads of the two viruses has converged. Between 48 and 120 h, Pang2017 did not proliferate noticeably, whereas XBB.1.16 increased by approximately 0.9 log.

In Figure 6, the results from the plaque assay indicated that the size of plaques formed by Pang2017 was about 60% of those formed by XBB.1.16. This suggests that Pang2017’s invasive ability in cells is slightly inferior to that of XBB.1.16 [63,64,65]. As seen in Figure 6, Pang2017 and XBB.1.16 exhibit similar growth characteristics in Vero cells. However, in the later stages of infection (after 48 h), Pang2017’s cytotoxicity and proliferation ability are slightly lower than those of XBB.1.16. XBB.1.16 is a currently prevalent SARS-CoV-2 variant, and compared to the wild-type strain, its pathogenicity is considered reduced. On the other hand, Pang2017 is not a human pathogen and is highly attenuated in animals. The biological comparative analysis of the two viruses aligns with the known virulence of each virus.

## 3. Discussion

### 3.1. COVID-19 Special Relationship with Pangolin-CoVs: Can Be Found in the Abnormally Hard M: Burrowing Animal

The fact that pangolin-CoVs have generally an approximately 90% genetic similarity to SARS-CoV-2, as compared to RaTG13 and BANAL-236 of about 96%, causes many scientists to jump to the conclusion that pangolin-CoVs are not that closely related to SARS-CoV-2, not to mention being a recent ancestor. Such a conjecture, however, hides the intricacies of genetics and evolution, especially with regard to the likelihood of the presence of recombination [56], which inevitably leads to gross errors. Indeed, at least one laboratory has noticed that portions of the S proteins of pangolin-CoVs resemble most closely those of the SARS-CoV-2 S protein. In the present study, a complex computational and experimental approach to looking at different viral proteins is used to study this enigma.

The computational side of our study that involves SDMs has uncovered a specific feature seldom found in CoVs, with the exception of CoVs of burrowing animals, such as rabbit-CoV (Figure 1). This tell-tale feature has to do with the fact that SDMs have uncovered that all thus far known COVID-19-related viruses have a hard outer shell (low PID_M_). The presence of this unique and peculiar property is corroborated independently by another laboratory that showed that SARS-CoV-2 is much more persistent in environments away from light than all tested CoVs [66].

The presence of this hard outer shell presents a few intriguing questions, such as why is this even present? and how does this affect the transmissibility of the virus? The hallmark hard M arises from SARS-CoV-2 close evolutionary relationship with a burrowing animal, the pangolin. CoVs of burrowing animals need to rely on feces that have been buried for a long time to facilitate their transmission [15,16,32,37]. A hard outer shell, and sometimes even an inner shell, is required for the virus to stay viably active in the feces for a long time, as the outer and, to some extent, inner shells protect the virion from damage [14,15,16,17,18,19,44,45,46,47,48]. It is therefore not surprising that SARS-CoV-2 has been experimentally shown to be more resilient away from light than the available referenced CoVs used as controls [66].

### 3.2. Evidence of an Even Closer Relationship between Omicron and Pangolin-CoVs

The harder outer (lower PID_M_) and inner (lower PID_N_) shells result, however, in different manifestations of the disease caused by the virus. A harder N (lower N disorder) causes greater attenuation of the virus, as N is intimately involved in the replication process, whereas a harder M provides greater resistance to the anti-microbial enzymes found in saliva and mucus. This is the reason why SARS-CoV-2 was highly infectious, unlike SARS-CoV-1. This SDM prediction has been reproduced by Wolfel et al. [49], who clinically showed that COVID-19 patients shed much larger amounts of viral particles than 2003 SARS-CoV-1 patients. Furthermore, the shared rigidity of M among COVID-19 variants provides a unique opportunity for a glimpse into the evolution of SARS-CoV-2 via phylogenetic studies, because phylogenetic studies are often affected by the potential errors in the presence of recombination, but M, on the other hand, may be able to alleviate this problem because it is likely to be very conserved given its rigidity. Indeed, the phylogenetic trees created using M have been somewhat different from the trees created using other CoV proteins or the entire genome. As seen in Figure 3A, our phylogenetic tree places SARS-CoV-2 (SARS2) in close proximity to Pang2019, thereby suggesting a very close evolutionary relationship between pangolin-CoVs and SARS-CoV-2.

An even more intriguing picture is seen in Figure 3B, showing that Omicron is more closely related to bat-CoVs and pangolin-CoVs than the other SARS-CoV-2 variants. How could this be so? A hint to the answer is likely to involve an already-mentioned mystery pertaining to the fact that Omicron has been found to be very genetically different from all previous variants [35]. This led some scientists to believe that Omicron had been hiding in an animal host, such as mice [67]. According to SDMs, mice are theoretically a possibility, but mice have dual evolutions depending on the species. Mice and rats in urban settings have evolved to live in the homes of humans since the dawn of civilization, but those in rural environments live in burrows [68]. We have therefore reasons to believe that mice and rats, unlike pangolins, may not provide the most ideal environments for the hardening of M and N, even if the scenario is possible according to SDMs.

While the phylogenetic tree in Figure 3A,B using Clustal Omega [57,69] places SARS-CoV-2 and Omicron in very close proximity to pangolin-CoVs and pangolin-CoVs/bat-CoVs respectively, another M phylogenetic study using CLUSTALW (TREX: CLUSTALW (http://www.trex.uqam.ca/index.php?action=align&project=trex, accessed on 8 June 2024) [19,32,58,70] with distance optimization has placed Omicron even closer to pangolin-CoVs. Phylogenetic studies using M are not the only evidence of a closer relationship between Omicron and pangolin-CoVs. For instance, Table 2 shows that the original Omicron subvariant, BA.1, has an even harder M (5.4%) than later XBB.1.16 (5.8%), and that the former has a much lower SARS-CoV-2 genetic similarity (98.7%) than the other variants (99.1%), which brings it closer to that of pangolin-CoVs (98.2%). Curiously, the later Omicron subvariant, XBB.1.16, has a PID_M_ (5.9%) and M genetic similarity (99.1%) that resemble the other SARS-CoV-2 variants, even though the phylogenetic study used M clusters XBB.1.16 and BA.1 together. Apparently, as time went on, Omicron began to pick up bits of genetic material from another variant, presumably Delta. Evidence of the even closer relationship between Omicron and pangolin-CoVs could suggest a greater probability that Omicron may have been hiding among pangolins before it surfaced among humans in South Africa.

### 3.3. Range of SARS-CoV-2 N Disorder Matches That Pangolin-CoVs2017, Not Bat-CoVs

Yet another piece of evidence that depicts the close relationship between pangolin-CoVs and SARS-CoV-2/Omicron can be found when we inspect the peculiarities of disorder distribution in N. The pattern of N disorder in pangolin-CoVs closely matches that of SARS-CoV-2 in terms of range when you include Omicron. This is in sharp contrast to the COVID-19-related to bat-CoVs as seen in Table 3. We are able to see that both Omicron and pangolin-CoVs are able to reach a lower threshold of 44.5–44.8, which is not seen in bat-CoVs. The reason for the larger range of PID_N_ values of pangolin-CoV can be traced to its fecal-oral transmission route. As mentioned, even though a hard outer shell is essential to protect the virus from a harsh environment, a harder inner shell (lower N disorder) often adds additional protection. This has also been seen in many other viruses, including rabies and DENV [14,17,18]. We see, however, the COVID-19-related bat-CoV PID_N_ remaining within the range seen in bat-CoVs in general, as bats are flying animals that require higher N PIDs for respiratory viral transmission. Apparently, SARS-CoV-2, especially Omicron, was able to acquire its greater range of PID_N_, likely from its closer relationship with pangolin-CoV.

When the PID_N_ values of pangolin-CoVs were first calculated based on the available data, it became clear that the PID_N_ of Pang2017 was as low as 44.8%. For this reason, the SDMs predicted the Pang2017 to be attenuated and provided a window for a silent spread, as an attenuated virus can easily escape the attention of the medical community. Then came the Omicron outbreak, which first started in South Africa. Unlike previous variants, Omicron PID_N_ (BA1, 44.7%) was found to be quite similar to Pang2017 (44.8%), and, unsurprisingly, Omicron was clinically and experimentally found to be milder than previous variants. Similarly, at least two laboratories have experimentally shown that Pang2017 is attenuated. In agreement with SDMs, as seen in Table 3, no such attenuation was seen in BANAL-236 (PID_N_: 48.6%) and Pang2019 (PID_N_: 48.6%), which were instead shown to be even more aggressive than Wuhan-Hu-1 in certain cells.

### 3.4. Differences in Pang2017 and XBB.1.16 N Disorder Patterns Can Explain Subtle Discrepancy in Experimental Results for the Two Viruses

While viral titrations have been determined for Pang2017, Pang2019, SARS-CoV-1, and BANAL-236 in comparison to non-Omicron SARS-CoV-2, none has yet to be determined for Pang2017 with Omicron until now with this paper. While Figure 6 shows that XBB.1.16 and Pang2017 have similar cytopathic effects, viral growth, and plaque formation, even if there are subtle differences. SDMs attribute the similarities to the similarity in PID_N_ values (Pang2017 PID_N_: 44.7%, XBB PID_N_: 44.5%); a more careful study using disorder and molecular analysis is needed to understand the subtle differences. The PONDR^®^ VLXT plots in Figure 5 tell us that even though Pang2017 has a slight advantage in N disorder (44.7%), much of the greater disorder lies in the NLS region, not within the NTD RNA-binding region. This basically means that Pang2017 N is more efficient in entering the nucleus of the host cell to change the cell cycle to its advantage, whereas XBB.1.16 N has greater ability to bind to RNA. These properties can be seen in the viral titration curve. The higher RNA copies of Pang2017 seen at 9–48 h after initial infection are likely the result of the greater ability of Pang2019 N to enter the nucleus to delay the cell cycle, while the higher disorder at the NTD RNA-binding region of XBB.1.16 N allows for slightly greater viral replication after 48 h. We need to remember that all SARS-CoV-2 variants, including XBB, have a furin cleavage site (FCS) that is missing in all pangolin-CoVs and COVID-19-related bat-CoVs. For this reason, the presence of FCS in XBB.1.16 but not in Pang2019 is likely to add to the slightly higher efficiency of the former. Even so, it is important to note that FCS is not likely to have given much advantage given the result we have. This note may be important given the debate on the importance of FCS in infectiousness. The disorder patterns detected using molecular analysis can be used for further attenuation, which will be useful for future vaccine development.

## 4. Materials and Methods

### 4.1. Computational Biology: SDMs and Protein Intrinsic Disorder

The set of Shell Disorder Models (SDMs) consists of three closely related models that measure the intrinsic disorder levels in the shell proteins of various viruses in order to predict their properties, such as spread and virulence. SDMs use the PONDR^®^-VLXT (www.pondr.com, accessed on 8 June 2024) [13,14,71] algorithm to determine the levels of predicted order in proteins. Protein intrinsic disorder is defined as an entire protein or a portion of a protein that has no unique 3D structure. Disorder allows for many important protein roles, such as more efficient molecular, recognition in protein-protein/RNA/glycoprotein/DNA/lipid binding [28,33,34,35,36,37,59,60]. PONDR^®^-VLXT is a neural network that is trained using known sequences of proteins that are ordered and disordered [13,14,72]. It has been successfully used to study a large number of viruses, including Dengue virus (DENV), Ebola virus (EBOV), Nipah virus (NiV), HIV, influenza virulence, MERS-CoV, SARS-CoV-1, and SARS-CoV-2 [14,15,16,17,18,19,20,32,42]. PONDR^®^-VLXT predicts if each residue is ordered or disordered, given the sequence of a protein. An important ratio that will be used extensively to determine the level of disorder of a protein is the percentage of intrinsic disorder (PID), which is defined as the number of disordered residues divided by the total number of residues multiplied by 100. Protein sequences for these analyses were taken from NCBI-Protein [73,74] and UniProt [75].

The database containing information on disorders and sequences of viral proteins was stored in MYSQL [76] using JAVA [77] and Python [78]. The phylogenetic trees were constructed using CLAUSTALW and CLUSTAL Omega, which are available at the TREX and EMBL-EBI websites, respectively [57,58,69,70]. The PONDR^®^-VLXT plots were generated using OpenOffice Spreadsheet [79] and adapted for publication using GIMP [71]. Statistical analyses were performed using the R package [80,81]. NCBI BLASTP (https://blast.ncbi.nlm.nih.gov/Blast.cgi?PAGE=Proteins, accessed on 8 June 2024) [82,83] was used to determine the genetic similarities of proteins. The corresponding schematic diagrams were drawn using both OpenOffice [79] and GIMP [71].

### 4.2. Experimental Biology: Cells and Viruses

SARS-CoV-2 XBB.1.16 isolate/2023 was passaged three times in Vero cells (ATCC, CCL-81), and Pang2017 (previously named GX_P2V) infectivity titer was determined by plaque assay. The Pangolin coronavirus was derived from a plaque clone of the early culture of Pang2017. This clone has a truncated 3’-UTR mutation and exhibits cross-neutralization epitopes with SARS-CoV-2 [22,84]. Vero cells were purchased and cultured in Minimum Essential Medium (MEM, Gibco) supplemented with 10% *vol*/*vol* FBS (PAN) and 1% *vol*/*vol* Anti-Anti (Gibco). Viral infection was conducted by first removing the culture medium, then adding the virus stock at a multiplicity of infection (MOI) of 0.01, and rocking at room temperature for 2 h. After two washes with phosphate-buffered saline (PBS, Hyclone), the cells were further incubated in MEM supplemented with 2% *vol*/*vol* FBS (PAN) and 1% *vol*/*vol* Anti-Anti (Gibco) at 37 °C in a 5% CO_2_ incubator. After 48 h, the culture supernatant was harvested, aliquoted, and stored at −80 °C.

### 4.3. Experimental Biology: Viral One-Step Growth Curve

A day prior to infection, Vero cells were diluted in MEM complete medium at a density of 3.3 × 10^5^ Vero/mL and seeded at 3 mL/well in six-well plates. Cells in the six-well plates were infected with virus stocks at MOI 0.1 and incubated at room temperature, with even distribution ensured by rocking. After thorough washing with PBS, cells were maintained in MEM supplemented with 2% *vol*/*vol* FBS and 1% *vol*/*vol* Anti-Anti and incubated at 37 °C in a 5% CO_2_ cell incubator (PHCbi). At the indicated time points (2, 4, 6, 8, 10, 12, 24, 36, 48, 72, and 120 h post-infection), 200 µL of the culture supernatant was collected and stored at −80 °C. The lost volume was replenished with fresh pre-warmed medium with a consistent serum concentration of 2% FBS. This experiment was conducted in triplicate, and viral RNA in the supernatant was quantified by the following RT-qPCR.

Specifically, viral RNA was extracted using the cell/tissue total RNA extraction kit (Nobelab, China) as per the manufacturer’s instructions and reverse transcribed using the HiScript III RT SuperMix for qPCR (+gDNA wiper) kit (Vazyme, China). Quantitative PCR targeting the viral N gene was performed using the QuantiNova Probe PCR Kit (QIAGEN, Germany). The primer and probe sequences for Pang2017 detection were: forward primer: TCTTCCTGCTGCAGATTTGGAT, reverse primer: TTACACATTAGGGCTCTTCCATATAGG, and probe: FAM-TGCAGACCACACAAGGCAGATGGGC-MGB. The primer and probe sequences for XBB.1.16 detection were: forward primer: TCTTCCTGCTGCAGATTTGGAT, reverse primer: ATTCTGCACAAGAGTAGACTATATATCGT, and probe: FAM-TGCAGACCACACAAGGCAGATGGGC-MGB. Following the manufacturer’s instructions, primers and probes were used at final concentrations of 200 nM and 100 nM, respectively. Plasmids containing amplification fragments of Pang2017 and XBB.1.16 were used as standards for quantifying the N gene of the two viruses.

### 4.4. Experimental Biology: Cytopathic Effect Analysis (CPE) and Plaque Assay

For CPE analysis, the virus infection procedure was the same as described above. At designated time points, cell images were captured under an optical microscope (Olympus, Japan) using FCSnap software at a fixed magnification (100×). Plaque assay was conducted using methylcellulose (a semi-solid, low viscosity overlay) to determine plaque size and plaque-forming unit titers, as previously described [22]. Briefly, Vero cells were seeded in six-well plates, and upon reaching full confluence, they were infected with serial dilutions of virus stock in virus culture medium with 2% FBS. Each well received 500 µL of the diluted virus, which was rocked at room temperature for 2 h. Subsequently, the infection medium was discarded, cells were washed with PBS, and 3 mL of 1% wt/vol methylcellulose (prepared by mixing 2 × MEM with 2% methylcellulose at a 1:1 ratio, with FBS and Anti-Anti adjusted to 2% and 1% vol/vol, respectively) was added. After incubating at 37 °C in a 5% CO_2_ incubator for 72 h, the cells were fixed and stained for the plaque analyses.

## 5. Summary and Conclusions

The evolutionary footprint of pangolin-CoVs can be found in all COVID-19-related viruses in the form of a hard outer shell (abnormally low PID_M_ values), which is a necessity for viral transmission via the buried feces of a burrowing animal, the pangolin. The fact that all thus far known COVID-19-related SARS-CoV-2 variants have abnormally hard M, seldom seen in any CoV with the exception of CoVs related to a burrowing animal such as rabbits, is itself evidence that not only do pangolin-CoVs play a major evolutionary role in all COVID-19-related viruses, but also pangolins are likely to serve as a major reservoir for COVID-19-related viruses. Bat-CoVs, such as RaTG13 and BANAL, have the highest genetic similarities to SARS-CoV-2 at approximately 96%, in contrast to the approximately 90% in Pangolin-CoVs. To jump to the conclusion, however, that SARS-CoV-2 came directly from bats would be to ignore the probability that SARS-CoV-2 had been moving to and fro between pangolins and humans for a long time. We have further evidence of this. The seemingly greater genetic difference in the pangolin-CoVs thus found may be a reflection of the greater diversity of pangolin-CoVs. In fact, the diversity in N disorder of SARS-CoV-2 more closely resembles that of pangolin-CoVs, not bat-CoVs. The emergence of Omicron is highly enigmatic, as it is genetically very different from the other SARS-CoV-2 variants. Phylogenetic studies using M under two different algorithms show that Omicron, including XBB.1.16, is closer to pangolin-CoVs than the other variants. This provides further evidence that Omicron could have been hiding in pangolin all the while. Yet further suggestion that Omicron could have been hiding in a burrowing animal is seen by two factors. The first is the observation that came when it was noticed that the early Omicron subvariant BA.1 has an even harder outer shell than the other SARS-CoV-2 (PID_M_: 5.4% versus 5.9%).

Where did Omicron acquire its harder outer shell? A likely scenario is that it has been hiding in a burrowing animal such as a pangolin. Viruses of a burrowing animal host need a harder outer shell to last a longer time in buried feces in order to be viable to spread. It has also been known from other viruses that, while not necessary, a harder inner shell also helps protect the virus from environmental damage. For these reasons, it is therefore not surprising to observe that Omicron also ha slower PID_N_ values (PID_N_: 44.7%, BA.1, 44.5%, XBB.1.16, 48.2%, Wuhan-Hu-1) than the other variants. Again, pangolin is likely to create a better environment for the evolutionary hardening of both M and N as a burrowing animal, which is the reason that pangolin-CoVs have a greater range of PID_N_ values that match those of SARS-CoV-2. A lower PID_N_, however, has important implications. Like many viral inner shell proteins, it is intimately associated with the replication process, and a lower N disorder of Pang2017 and Omicron allows SDMs to predict attenuation in both viruses, which have similar PID_N_ values (pang2017: 44.8%, XBB.1.16: 44.5%). Our experimental study shows that both viruses exhibit similar properties, such as growth rates in viral titration and plaque sizes. There are, of course, subtle differences, but these subtle differences can be traced to the differences in the localities of the disordered regions of the two viruses and the presence of FCS in Omicron. The molecular analysis of N using disorder that is able to account for the small discrepancies can be further utilized. The molecular analytical methodology using disorder that is able to detect the presence of a greater disorder at the NTD RNA-binding region of XBB.1.16, responsible for its greater proliferation at a later stage, can be used to design further attenuation in future vaccine development.

## Figures and Tables

**Figure 1 ijms-25-07537-f001:**
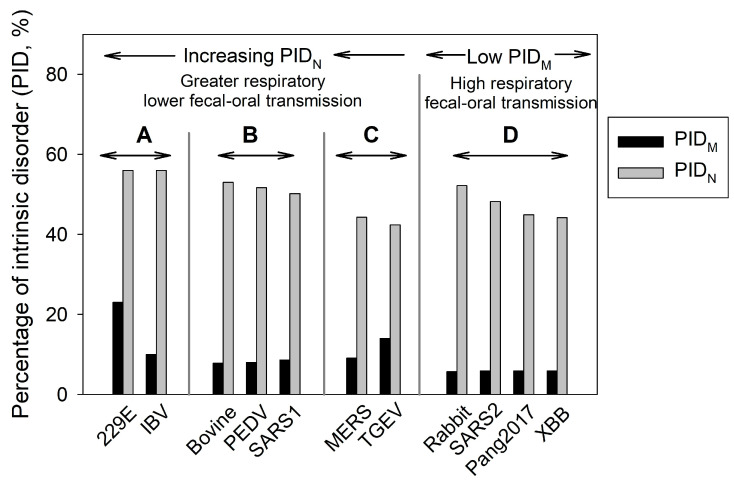
CoV Transmission SDM. Groups A–C are heavily dependent on PID_N_, whereas CoVs in group D have abnormally low M disorder (PID_M_), which includes all COVID-19-related viruses. (r = 0.78, *p* < 0.05).

**Figure 2 ijms-25-07537-f002:**
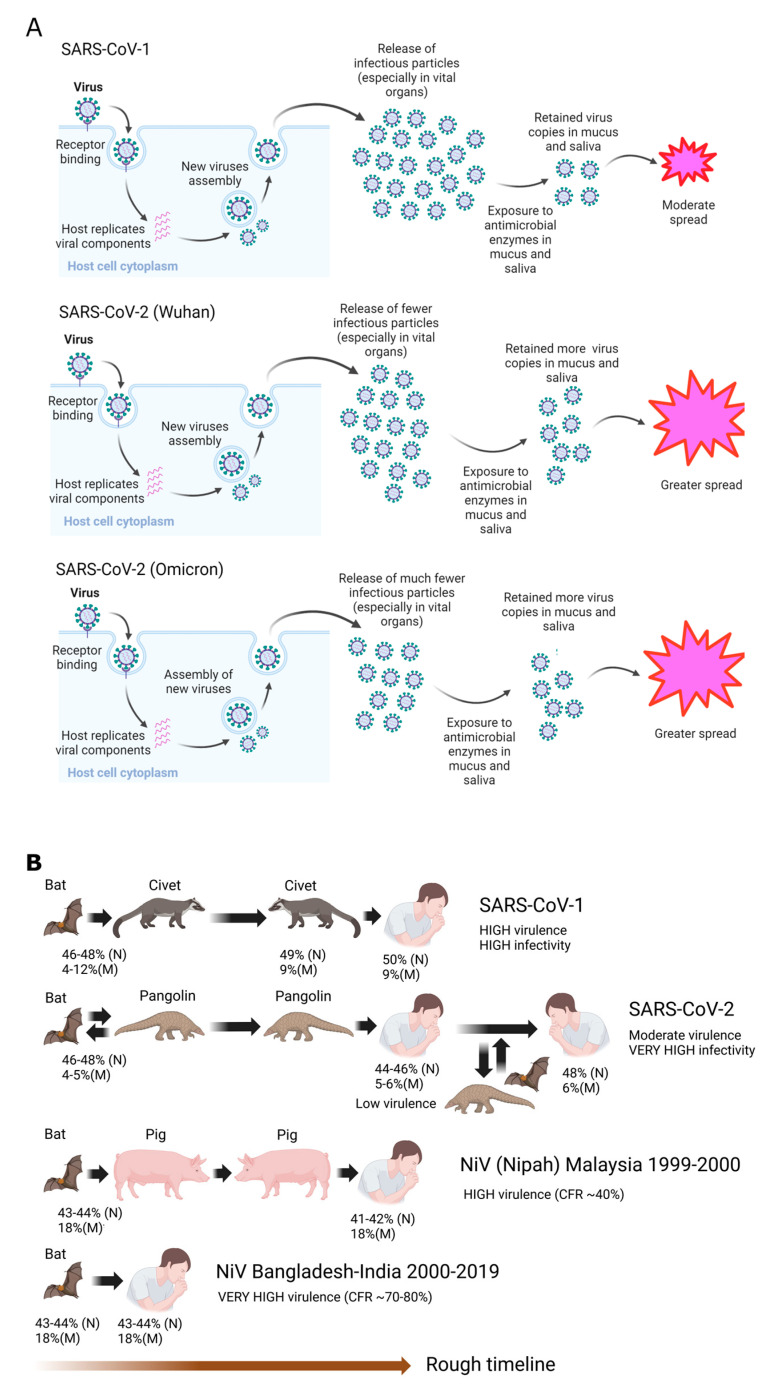
SDMs: Effects of Evolutionary Mutations on the Shell Proteins. (**A**) A harder M protects the virion from damage incurred from antimicrobial enzymes in the saliva and mucus, as in the case of SARS-CoV-2, even if a more disordered N allows replication of more infectious particles, especially in vital organs, which is the case in SARS-CoV-1. (**B**) Attenuation is likely to occur when the virus enters animal hosts like pigs or pangolins, where fecal-oral transmission is an important route. It is believed that SARS-CoV-2 entered and re-entered pangolins multiple times before it appeared as Wuhan-Hu-1 or Omicron.

**Figure 3 ijms-25-07537-f003:**
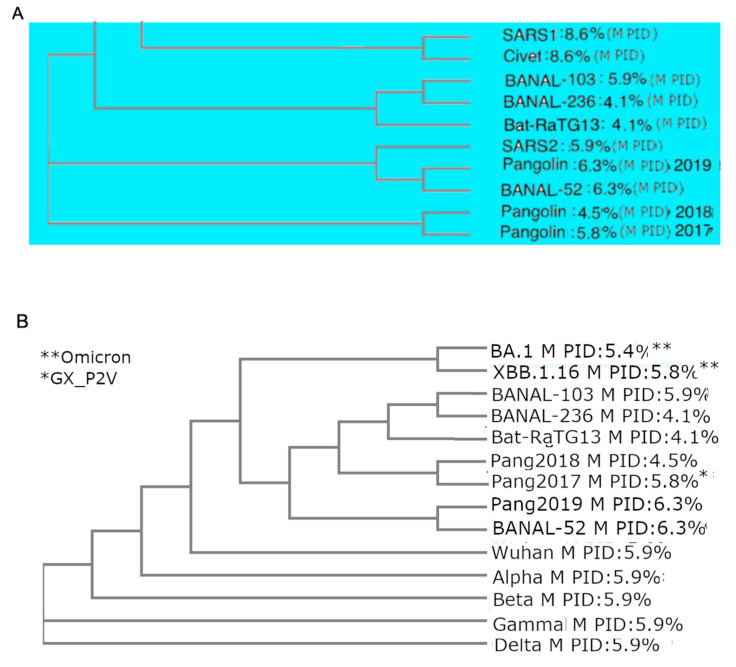
CoV Phylogenetic Trees Using M protein. Omicron (XBB.1.16 and BA.1) is clustered alongside pangolin-CoVs and bat-CoVs, not SARS-CoV-2. (**A**) Tree that includes SARS-CoV-1 (SARS1) and COVID-19-related viruses. (**B**) Tree that includes COVID-19-related viruses and SARS-CoV-2 variants. Both trees were generated using CLUSTAL OMEGA [57]. A similar tree using M but utilizing CLUSTALW [58] with distance optimization placed Omicron even closer to pangolin-CoV [19,32].

**Figure 4 ijms-25-07537-f004:**
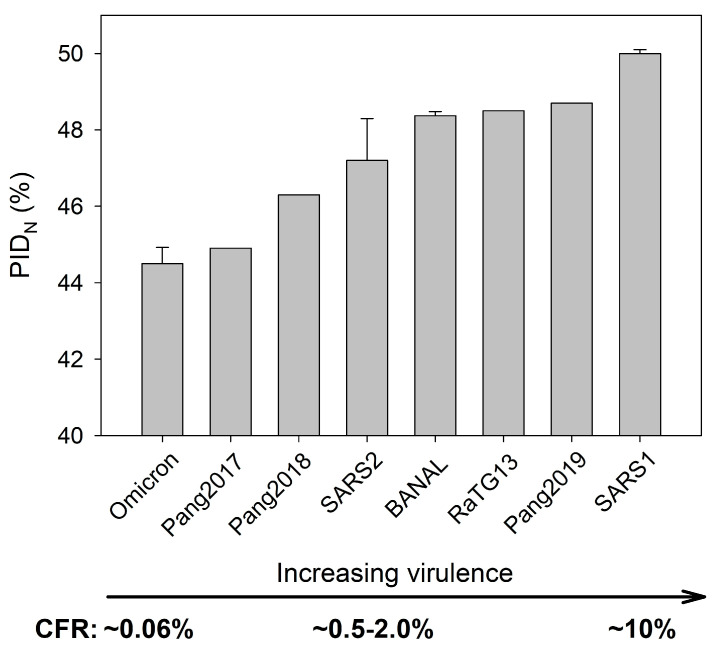
Virulence-Inner Shell Disorder Model as Applied to COVID-19-Related Viruses. Using mutivariate analysis, a strong correlation (r = 0.8) between SARS-CoV-2, Omicron PID_N_, and CFR has been found. CFRs of SARS-CoV-2 (SARS2, non-Omicron) have been estimated to be 2.0–0.5%. CFRs of various SARS-CoV-2 viruses and SARS-CoV-1 viruses were collected from various published sources [14,24,25], whereas PID_M_ and PID_N_ were calculated from PONDR^®^-VLXT using sequences as seen in Table 2.

**Figure 5 ijms-25-07537-f005:**
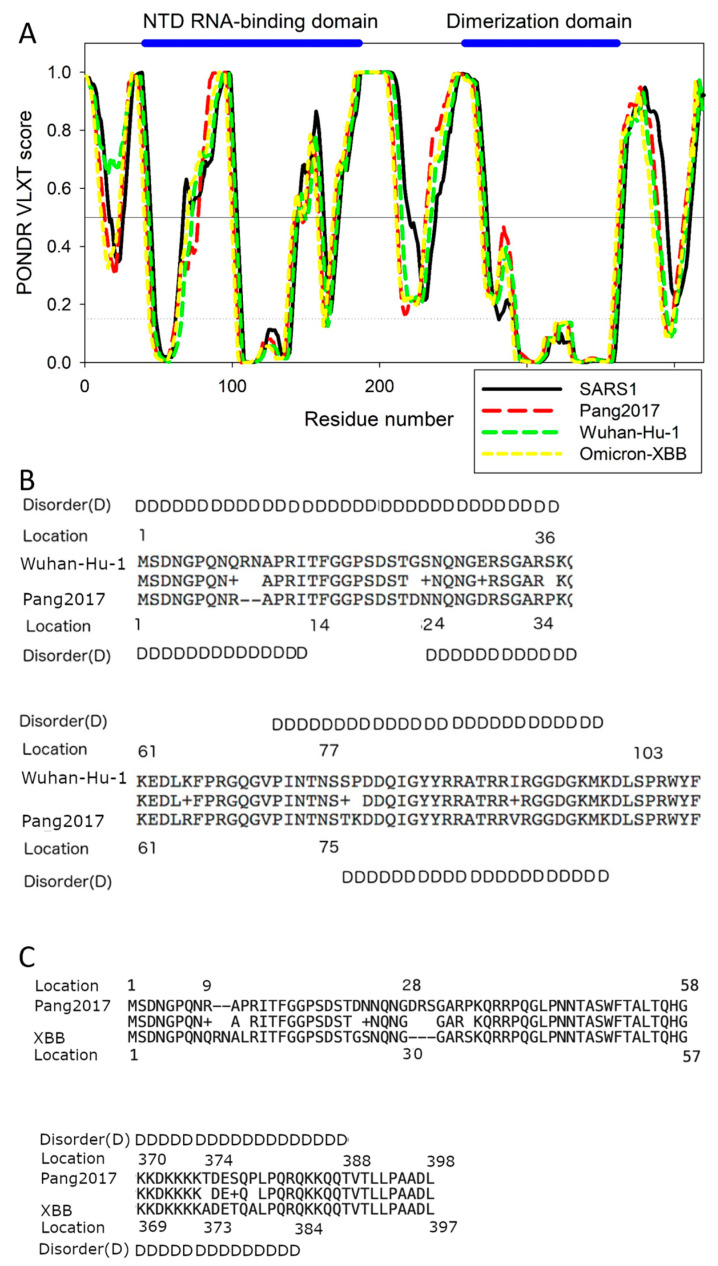
PONDR^®^-VLXT Plots and Sequence Comparison. (**A**) PONDR^®^ VLXT plots for the comparison N of SARS-CoV-1 (SARS1). (**B**) Sequence and disorder analysis of crucial N regions of Wuhan-Hu-1 and Pang2017. (**C**) Comparison of crucial N regions of Pang2017 and Omicron XBB (XBB) using sequence and disorder analysis. Residues predicted to be disordered are denoted by “D” in (**A**,**B**).

**Figure 6 ijms-25-07537-f006:**
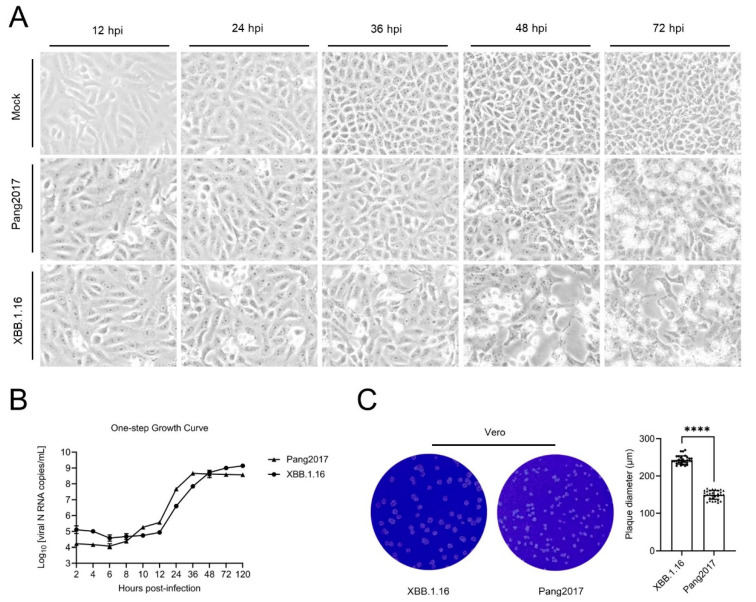
Comparative Analysis of Pangolin Coronavirus Pang2017 and SARS-CoV-2 XBB.1.16 Growth Characteristics in Vero Cells. (**A**) Microscopic evaluation of cytopathic effects in Vero cells post infection with Pang2017 and XBB.1.16 reveals analogous cellular morphological changes, including cell degeneration and rounding. By 48 h, XBB.1.16-infected cells display pronounced monolayer disruption; (**B**) one-step growth kinetics of Pang2017 and XBB.1.16 in Vero cells over time. Pang2017’s viral genome exhibited an initial increase between 8–10 h post-infection, with a consistently higher concentration relative to XBB.1.16 from 10–36 h. Following 48 h post-infection, Pang2017’s proliferation plateaued, lagging behind XBB.1.16. (**C**) At the 120-h mark, plaque assays demonstrated that Pang2017 (149.4 ± 11.85 µm, n = 30) forms significantly smaller plaques than XBB.1.16 (242.7 ± 11.14 µm, n = 30) in Vero cells. Data are presented as means ± SD. **** *p* < 0.0001, determined using a two-tailed Student’s *t*-test.

**Table 1 ijms-25-07537-t001:** Shell Disorder Models (SDMs). Three closely related SDMs were built based on levels of disorder in the shell proteins of viruses and known characteristics of viruses.

Year of First Publication	Shell Disorder Model	Details
2008	Parent Viral Shape-shifter Model	Abnormally huge levels of disorder were found at the outer shell of many HIV-1 variants and may be sexually transmitted viruses such as HSV and HCV. This could account for the lack of an effective HIV vaccine.
2012	CoV Transmission SDM	Levels of fecal-oral and respiratory CoV transmission are predicted by levels of shell disorder.
2015	Virulence-inner Shell Disorder Model	High correlations between the inner shell and the virulence of a variety of viruses have been detected.

**Table 2 ijms-25-07537-t002:** Bat-CoVs, COVID-19-related viruses and SARS-CoV-1 (2003 SARS-CoV). All known COVID-19 viruses have abnormally hard outer shells (low PID_M_ values). The pattern of N PID for pangolin-CoVs is more similar to that of SARS-CoV-2 and Omicron, not bat-CoVs. Non-COVID-19-related bat-CoVs are listed here for comparison purposes. SDMs have detected two sub-variants of Delta, not previously observed. We named them Delta1 and Delta2.

Coronavirus	Sequence Similarity M (%)	PID_M_ (%)	Accession: UniProt(U) GenBank(G)	Sequence Similarity N (%)	PID_N_ (%)	Accession: UniProt(U) GenBank(G)
SARS-CoV-1	90.5	8.6	P59596(U)	90.5	50.2	P59595(U)
Civet-SARS-CoV	90.1	8.6	Q3ZTE9(U)	90.01	49.1	Q3ZTE4(U)
COVID-Related Bat-CoVs		6.0 ± 0.2			48.3 ± 0.2	
RaTG13	99.6	4.1	QHR63303(G)	99.1	48.5	QHR63308(G)
Laotian Bat-CoV						
[Banal-52]	98.7	6.3	UAY13220(G)	99.3	48.5	UAY13225.1
[Banal-103]	98.7	5.9	UAY13232(G)	99.1	48.5	UAY13257.1
[Banal-236]	99.1	4.1	UAY13256(G)	99.3	48.2	UAY1326.1
Pangolin-CoV		5.6 ± 0.9 ^a^			46.6 ± 1.6 ^a^	
2019	98.2	6.3	QIG55948(G)	98	48.7	QIG55953(G)
2018	97.7	4.5	QIQ54051(G)	93.8	46.3	QIQ54056(G)
**2017 *****	**98.2**	**5.9**	**QIA48617(G)**	**94**	**44.8**	**QIA48630(G)**
				93.32	46.5	QIA48656(G)
SARS-CoV-2						
[Wuhan]	100	5.9	YP009724393(G)	100	48.2	YP009724397(G)
[Delta1]	99.1	5.9	QUX81285(G)	99.3	46.8	QYM89997(G)
[Delta2]	99.1	5.9	QUX81285(G)	99.1	47.5	QYM89845(G)
**[Omicron]** ***	**98.7**	**5.7 ± 0.4**		**98.6**	**44.5 ± 0.4**	
Omicron						
BA.1.44	98.7	5.4	UFO69282(G)	98.6	44.7	UFO69287(G)
XBB.1.16	99.1	5.9	WIL50320(G)	98.3	44.2	WIL50325(G)
Bat-CoVs		11 ± 15 ^a^			47.7 ± 0.9 ^a^	
RATG13	99.6	4.1	QHR63303(G)	99.1	48.5	QHR63308(G)
Bat 512	35.5	15.3	Q0Q463(U)	29.4	46.5	Q0Q462(U)
HKU3	91	7.7	Q3LZX9(U)	89.6	48	Q3LZX4(U)
HKU4	42.7	16.4	A3EXA0(U)	51.1	48.5	A3EXA1(U)
HKU5	44.7	11.8	A3EXD6(U)	47.9	47.1	A3EXD7(U)

^a^ Standard deviation is denoted by “±”. *** Attenuated strains were detected.

**Table 3 ijms-25-07537-t003:** The Correlation (r = 0.8) between PID_N_ and Viral Growth.

Virus/Isolate	PID_N_	Non-Attenuation/Aggressiveness
SARS-CoV-1	50.00%	+++
BANAL-236	48.5%	++
Pang2019	48.5	++
Wuhan-Hu-1	48.20%	+
XBB.1.16	44.5	-

“+” and “-” denote the levels of aggressiveness. For example, “-”, “+” and “++” represent “less aggressive”, “aggressive” and “more aggressive” respectively.

## Data Availability

Data are contained within the article.

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
