# Peer review of "A Comparative Experimental and Computational Study on the Nature of the Pangolin-CoV and COVID-19 Omicron"

_ijms, 2024, doi:10.3390/ijms25147537_

Round 1

Reviewer 1 Report

Comments and Suggestions for Authors

Wei L. et al. using comparative experimental and computational approaches have studied the nature of pangolin-CoV and SARS-CoV-2. The manuscript is interesting and well-written, however, a few issues should be resolved.

Minor points:

  1. The ATCC No. of Vero cells should be added to the manuscript.
  2. Figure 2 description should be corrected, the A and B letters should be on the same level.
  3. Figures 3 and 5 need proper description, there is no reference associated with A, B, and C letters.
  4. Table 2 includes a few mistakes that should be corrected, e.g., 98,7, 44.5+0.4.
  5. There are a few mistakes associated with the CoV nomenclature, e.g., line 356 - SAR-CoV-2, line 622 - SARS_CoV-1.
  6. Line 578, the proper name of the ClustalW program should be included in the manuscript.
  7. The figures' legends should be consistent in the whole manuscript, e.g., A or (a).
  8. Line 678, the bracket after the SARS-CoV-2 should be removed.
Comments on the Quality of English Language

Minor editing of English language required.

Author Response

Reviewer #1

Wei L. et al. using comparative experimental and computational approaches have studied the nature of pangolin-CoV and SARS-CoV-2. The manuscript is interesting and well-written, however, a few issues should be resolved.

We would like to thank the reviewer for his/her highly insightful comments and we have tried our best to address his/er concerns. Please also note that the entire Material and Methods section has been moved to after the Discussion section as requested by the Editor.

Minor points:

  1. The ATCC No. of Vero cells should be added to the manuscript.

    Changed accordingly. Line 128: “SARS-CoV-2 XBB.1.16 isolate/2023 was passaged three times in Vero cells (ATCC, CCL-81) …”

  2. Figure 2 description should be corrected, the A and B letters should be on the same level.

    Done. We have also placed Figure 2A and Figure 2B vertically, instead of horizontally, for greater image clarity and enlargement.

  3. Figures 3 and 5 need proper description, there is no reference associated with A, B, and C letters.

    Done. Changes made:

    "Figure 3. CoV Phylogenetic Trees Using M protein. Omicron (XBB.1.16 and BA.1) is clustered alongside pangolin-CoVs and bat-CoVs, not SARS-CoV-2. A. Tree that includes SARS-CoV-1 (SARS1) and COVID-19-related viruses. B. Tree that includes COVID-19-related viruses and SARS-CoV-2 variants. Both trees were generated using CLUSTAL OMEGA [57]. A similar tree using M but utilizing CLUSTALW [58] with distance optimization placed Omicron even closer to pangolin-CoV [26,30]".

"Figure 5. PONDR®-VLXT Plots and Sequence Comparison. A. PONDR® VLXT plots for the comparison N of SARS-CoV-1 (SARS1). B. Sequence and disorder analysis of crucial N regions of Wuhan-Hu-1 and Pang2017. C. Comparison of crucial N regions of pang2017 and Omicron XBB (XBB) using sequence and disorder analysis. Residues predicted to be disordered are denoted by “D” in A and B".

  1. Table 2 includes a few mistakes that should be corrected, e.g., 98,7, 44.5+0.4.

    Corrected

  2. There are a few mistakes associated with the CoV nomenclature, e.g., line 356 - SAR-CoV-2, line 622 - SARS_CoV-1.

    Corrected

  3. Line 578, the proper name of the ClustalW program should be included in the manuscript.

    We have done so but we added more as I have realized that we only gave the reference for ClustalW but not ClustalW Omega. We have corrected all these by placing both references. Thanks for bringing all these to our attention:

Changes made:

"Figure 3. CoV Phylogenetic Trees Using M protein. Omicron (XBB.1.16 and BA.1) is clustered alongside pangolin-CoVs and bat-CoVs, not SARS-CoV-2. A. Tree that includes SARS-CoV-1 (SARS1) and COVID-19-related viruses. B. Tree that includes COVID-19-related viruses and SARS-CoV-2 variants. Both trees were generated using CLUSTAL OMEGA [57]. A similar tree using M but utilizing CLUSTALW [58] with distance optimization placed Omicron even closer to pangolin-CoV [26,30]".

"While the phylogenetic tree in Figure 3 A-B using Clustal Omega [57] places SARS-CoV-2 and Omicron in very close proximity to pangolin-CoVs and pangolin-CoVs/bat-CoVs respectively, another M phylogenetic study using CLUSTALW (TREX: CLUSTALW (http://www.trex.uqam.ca/index.php?action=align&project=trex) [26,30,58] with distance optimization has placed Omicron even closer to pangolin-CoVs".

  1. The figures' legends should be consistent in the whole manuscript, e.g., A or (a).

    Corrected. We actually inadvertently placed two versions of Figure 6 with A-C and a-c on the previous version. We have corrected this. Thanks for bringing this to our attention.

  2. Line 678, the bracket after the SARS-CoV-2 should be removed.

Corrected

Reviewer 2 Report

Comments and Suggestions for Authors

The manuscript ''A Comparative Experimental and Computational Study on the 2 Nature of the Pangolin-CoV and COVID-19 Omicron'' present an interesting findings and provide an insight on relationship of omicron variant with Pangolin- CoV. 

Introduction of topic is satisfactory and well written and provide most of background information. 

The method section covered all information. 

The results finding explained here that greater disorder protein means more replication of virus (line 196-98), can authors explained it at cellular level or why this occurs? In line 205-236 can be improved at the moment it is bit confusing especially group A etc.

In line 244-45 authors claimed that SARS-CoV-2 is more pathogenic than SARS-CoV1 because of PIDn but there percentage is not much different (50% vs 48.2%) so how you want to support this findings or it indicates that PIDm is more important in pathogenicity? I guess this need to address as in further discussion hard outer shell (low PIDM) seems more important to establish relationship between omicron and other pangolin CoV. typographical error in line 245

Quality of figure 2A can be improved, also Figure 2 part A and B has been switched and need correction (line 264).

Some typographical error in naming CoV-2 (line 314). Table 2 need to correct for many typos error in naming CoV-2. 

Figure 3 has part A and B but in legend there is no information about separate parts so I guess it need to be done. 

I  guess figure 4 legends need more information about modelling of different viruses inner shell disorder. Line 361 mentioned about phylogenetic tree in figure 4 but figure 4 showed only PIDn/virulence information.

In figure 5c authors mentioned the plaque diameters of XBB.1.16 and pango2017 and have you counted number of plaques in both as in picture there is clear difference in number? Does difference in diameters is true indication of virulence of virus? 

Figure legend 6 is not correct and does not match with either results or picture.

In my opinion there is plenty of information in terms of harder shell of M or N and low/high PIDs among various strain of CoV and pango CoV, In my opinion that need correlation such as virulence or transmission in a summary form otherwise it is difficult to draw a proper conclusion., At the present all details in results and discussion need to give a clear indication of findings. 

Comments on the Quality of English Language

Quality of English language is satisfactory and may need some modification in results and discussion section.

Author Response

The manuscript ''A Comparative Experimental and Computational Study on the 2 Nature of the Pangolin-CoV and COVID-19 Omicron'' present an interesting findings and provide an insight on relationship of omicron variant with Pangolin- CoV. 

Introduction of topic is satisfactory and well written and provide most of background information. 

The method section covered all information. 

We would like to thank the reviewer for his/her highly insightful comments and we have tried our best to address his/er concerns. Please also note that the entire Material and Methods section has been moved to after the Discussion section as requested by the Editor.

The results finding explained here that greater disorder protein means more replication of virus (line 196-98), can authors explained it at cellular level or why this occurs? In line 205-236 can be improved at the moment it is bit confusing especially group A etc.

While we gave a brief description of these in viruses in general at that stage in the paper, we originally intended more specific details on CoVs later in the paper. Your point gave us the hint that we should at least provide a brief initial description on this. We agree:

In the case of CoVs, N plays a distinct role in the replication of the virus. An important role is the packaging of the infectious particles prior to their release. N has to wrap itself around the RNA and the greater N disorder provides for more efficient RNA-binding and, thus, more efficient assembly of the infectious particles that will later be released[43]. A second mode of action that N plays in the infectivity of the virus involves the entry of N into the nucleus of the cell in order to manipulate cell-cycle to the advantage of the virus. Greater N disorder could provide greater efficiency in its entry into the nucleus since greater disorder could enhance protein-protein and protein-lipid interactions. The domains in N that are responsible for the two modes of actions will be discussed in greater detail later in this paper”

As for the grouping of A B C and D, we added another paragraph. We believe the added paragraph may help resolve the confusion:

The initial model comprises groups A, B and C (see Figure 1), which are mainly based on PIDN , and, in fact, a strong correlation could be found between PIDN and the mode of transmission. M, on the other hand, poses an enigma at that time. A small but statistically significant correlation between PIDM and mode of transmission could be found[19.44]. The precise significance of this correlation was difficult to construe at that time. It was not until the arrival of COVID-19 with its incoming data, we were able to fully realized that the reason for our inability to further analyze the significance of PIDM then was due to the scarcity of data pertaining to CoVs with extremely low PIDM that are associated with burrowing animals [18,26,30]”.

In line 244-45 authors claimed that SARS-CoV-2 is more pathogenic than SARS-CoV1 because of PIDn but there percentage is not much different (50% vs 48.2%)

The small percentage difference (50 Vs 48.2) is both mathematically and biologically misleading. N and M are highly abundant proteins. M is the most abundant in the virion and N is the most abundant in the cell. N covers the entire RNA genome so as to protect the RNA from damage, whereas M spans the entire virion to protect the virion from damage. N is a huge protein of over 400 AA and 2% of 400 is 8 AA. This approximately ~8 AA mutations/disorder could be widespread through out N thereby affecting the function of the protein strongly. We must not forget also the abundance of N and M.

The percentage difference between 50% and 48.2 can be very misleading as N is a huge protein with over 400 amino acid residues and is a highly abundant protein both in the virion and cell [43.44]. Therefore, even a small difference in percentage could entail a fairly large number of mutations that involve widespread regions of disorder. This is also the case with M as M is somewhat smaller but is the most abundant protein in the virion [44]”

so how you want to support this findings or it indicates that PIDm is more important in pathogenicity? I guess this need to address as in further discussion hard outer shell (low PIDM) seems more important to establish relationship between omicron and other pangolin CoV.

Thanks for reminding us the need for such clarification. We need to remember that much of our work did not come out from nowhere but rather it came out of systematic studies of other viruses. Viruses have similar (not identical) proteins that play similar roles. We have strong statistical evidence that the inner shell disorder plays roles in the virulence of the virus in a large variety of unrelated viruses such as Nipah, Dengue and Ebola. We added the following paragraph as advised:

Their correlations with the modes of transmission were established using clinical and epidemiological data from farm animals, particularly porcine-CoVs. We need to keep in mind that much of the data available before 2003 were from the veterinary community, not medical community as CoVs were mainly associated with mild symptioms among humans before 2003, and they were not considered medically interesting [17,44,45] then”.

We have now seen that greater disorder in N (inner shell) can bring about greater virulence or greater infectivity depending on many factors such as the viral presence in vital organs, saliva or mucus. This idea is consistent not just in CoVs but also in other viruses we have studied in the past when it was found that inner shell of unrelated viruses such as DENV, EBOV and NiV [17,22,24,27,39], correlates with virulence. The outer shell (M, in the case of CoVs) is, however, more enigmatic. We don't have any evidence that the inner shell disorder is correlated with mortality rate of any virus. We have, however, some hints that the outer shell can be related to the ability of the virus' ability to penetrate vital organs and cause damage i.e. morbidity, not mortality, in cases such as Yellow Fever virus, Zika virus and HIV [17,21,22] . As already mentioned, in the case of CoVs, we have discovered that the hardness of M is related to the infectivity of COVID-19. We will also revisit this in Figure 2”.

We have never been able to find correlation between virulence and Outer shell disorder in any viruses including CoVs even though there are statistical hints of correlations between Morbidity, not Mortality, and M PID in Zika, Yellow Fever and HIV as a more disordered outer shell allows penetration to organs such as the placenta and brain and thereby causing damages.

typographical error in line 245

I am really sorry but I looked and re-looked but was unable to find the error. Maybe it is on a different line?

Quality of figure 2A can be improved, also Figure 2 part A and B has been switched and need correction (line 264).

We have placed 2A and 2B vertically so that Figure 2A can be enlarged and look better. I am not quite sure what the reviewer means by “switched”. I presumed he/she means the position of the letters “A” and “B” are not correct. If this is the case, we have corrected it.

Some typographical error in naming CoV-2 (line 314). Table 2 need to correct for many typos error in naming CoV-2. 

I am very sorry again. I am unable to find the typographical error. I think there is some confusion over its nomenclature. WHO naming convention has defined the virus as SARS-CoV-2 and its disease as COVID-19:

https://www.who.int/health-topics/coronavirus#tab=tab_1

We have however taken the liberty to name the 2003SARS-CoV as SARS-CoV-1 and to clarify this matter we placed “2003 SARS-CoV” in brackets after SARS-CoV-1. We have changed “bat-CoV” to “bat-CoVs”.

Figure 3 has part A and B but in legend there is no information about separate parts so I guess it need to be done. 

Done

Figure 3. CoV Phylogenetic Trees Using M protein. Omicron (XBB.1.16 and BA.1) is clustered alongside pangolin-CoVs and bat-CoVs, not SARS-CoV-2. A. Tree that includes SARS-CoV-1 (SARS1) and COVID-19-related viruses. B. Tree that includes COVID-19-related viruses and SARS-CoV-2 variants. Both trees were generated using CLUSTAL OMEGA [57]. A similar tree using M but utilizing CLUSTALW [58] with distance optimization placed Omicron even closer to pangolin-CoV [26,30]”.

Figure 5. PONDR®-VLXT Plots and Sequence Comparison. A. PONDR® VLXT plots for the comparison N of SARS-CoV-1 (SARS1). B. Sequence and disorder analysis of crucial N regions of Wuhan-Hu-1 and Pang2017. C. Comparison of crucial N regions of pang2017 and Omicron XBB (XBB) using sequence and disorder analysis. Residues predicted to be disordered are denoted by “D” in A and B”.

I  guess figure 4 legends need more information about modelling of different viruses inner shell disorder. Line 361 mentioned about phylogenetic tree in figure 4 but figure 4 showed only PIDn/virulence information.

In figure 5c authors mentioned the plaque diameters of XBB.1.16 and pango2017 and have you counted number of plaques in both as in picture there is clear difference in number? Does difference in diameters is true indication of virulence of virus? 

Thanks. We did not count the number of plaques, as this number depends on the viral titers added to the plates and does not provide meaningful information about viral virulence. In contrast, plaque size has traditionally been used to provide information regarding the replication kinetics and virulence of a virus (for example, PMID: 31579606, 35062259, and 28938842).

Figure legend 6 is not correct and does not match with either results or picture.

We have carefully revised the name of Pangolin coronavirus to Pang2017, and (a) to (A). We actually inadvertently placed two versions of Figure 6 with A-C and a-c on the previous version. We have corrected this. Thanks for bringing this to our attention.

In my opinion there is plenty of information in terms of harder shell of M or N and low/high PIDs among various strain of CoV and pango CoV, In my opinion that need correlation such as virulence or transmission in a summary form otherwise it is difficult to draw a proper conclusion., At the present all details in results and discussion need to give a clear indication of findings. 

There is much behind-the-scene collection of data in the effort to draw statistical conclusions. A full description is beyond the scope of this paper and could occupy an entire new paper. However, we can still try to address the concerns of the reviewer as best as we could.

We have forgotten to place the correlation number in Figure 2. We have corrected that (r=0.78, p < 0.05). Thanks for bringing this to our attention.

While the collection of M and N PIDs is straight-forward. The collections of infectivity and virulence are not. The collection of the transmissibility model came from clinical and epidemiological data from farm animals as described in our earlier papers because before 2003 most of the data pertaining to CoVs were related from the veterinary community, not medical aas before 2003 physicians were not interested in CoVs as they were associated with mild colds. The collection of CFRs can also be tricky as later CFRs may include vaccinated individuals. We had to be careful of such factors.

Changes are made to address the concerns:

Figure 4. Virulence-Inner Shell Disorder Model as Applied to COVID-19-Related Viruses. Using mutivariate analysis, strong correlation (r=0.8) between SARS-CoV-2 and Omicron PIDN and CFR has been found. CFRs of SARS-CoV-2 (SARS2, non-Omicron) have been estimated to be 2.0-0.5%. CFRs of various SARS-CoV-2 varaints and SARS-CoV-1 were collected from various published sources [36-38], wheras PIDM and PIDN were calculated from PONDR®-VLXT using sequences as seen in Table 2.

 The following passage is added

Their correlations with the modes of transmission were established using clinical and epidemiological data from farm animals, particularly porcine-CoVs. We need to keep in mind that much of the data available before 2003 were from the veterinary community, not medical community, as CoVs were mainly associated with mild symptoms among humans before 2003, and they were not considered medically interesting [17,44,45] then”.